# Dietary Guanidine Acetic Acid Addition Improved Carcass Quality with Less Back-Fat Thickness and Remarkably Increased Meat Protein Deposition in Rapid-Growing Lambs Fed Different Forage Types

**DOI:** 10.3390/foods12030641

**Published:** 2023-02-02

**Authors:** Wen-Juan Li, Yao-Wen Jiang, Zhao-Yang Cui, Qi-Chao Wu, Fan Zhang, He-Wei Chen, Yan-Lu Wang, Wei-Kang Wang, Liang-Kang Lv, Feng-Liang Xiong, Ying-Yi Liu, Ailiyasi Aisikaer, Sheng-Li Li, Yu-Kun Bo, Hong-Jian Yang

**Affiliations:** 1State Key Laboratory of Animal Nutrition, College of Animal Science and Technology, China Agricultural University, Beijing 100193, China; 2Zhangjiakou Animal Husbandry Technology Promotion Institution, Zhangjiakou 075000, China

**Keywords:** GAA, forage type, water-holding capacity, protein deposition, muscle gene expression

## Abstract

The aim of this study was to investigate whether guanidine acetic acid (GAA) yields a response in rapid-growing lambs depending on forage type. In this study, seventy-two small-tailed *Han* lambs (initial body weights = 12 ± 1.6 kg) were used in a 120-d feeding experiment after a 7-d adaptation period. A 2 × 3 factorial experimental feeding design was applied to the lambs, which were fed a total mixed ration with two forage types (OH: oaten hay; OHWS: oaten hay plus wheat silage) and three forms of additional GAA (GAA: 0 g/kg; UGAA: Uncoated GAA, 1 g/kg; CGAA: Coated GAA, 1 g/kg). The OH diet had a greater dry matter intake, average daily gain, and hot carcass weight than the OHWS diet. The GAA supplementation increased the final body weight, hot carcass weight, dressing percentage, and ribeye area in the longissimus lumborum. Meanwhile, it decreased backfat thickness and serum triglycerides. Dietary GAA decreased the acidity of the meat and elevated the water-holding capacity in mutton. In addition, the crude protein content in mutton increased with GAA addition. Dietary GAA (UGAA or CGAA) might be an effective additive in lamb fed by different forage types, as it has potential to improve growth performance and meat quality.

## 1. Introduction

In recent years, mutton consumption has continued to increase with the improvement of residents’ income level, but due to the long production cycle of sheep, the overall production efficiency is not high [1]. Hence, a question arises as to how to shorten the feeding cycle and increase the rapid-growing lambs’ meat production. The vitamins [2], amino acids [3], trace elements [4], and other nutritional additives have been widespread in husbandry to improve the growth performance and meat quality of animals in recent years in China.

A new nutritional feed additive, guanidine acetic acid (GAA) is a precursor of creatine, a conditionally essential nutrient that allows the storage of high-energy phosphate bonds and participates in energy metabolism and protein synthesis in muscle [5,6]. It is well known that GAA is mainly synthesized in the kidney by *L*-arginine: glycine amidinotransferase catalyzed from glycine and *L*-arginine. Guanidinoacetate methyltransferase (GAMT) then catalyzes the S-adenosylmethionine-dependent methylation of GAA to creatine. This process mainly takes place in the liver, kidney, pancreas, and testis [7]. It has been shown that GAA increase the creatine supply in Holstein heifers [8]. It has been reported that GAA or creatine supplement in the diet help to affect the muscle yield and meat quality of pigs, broilers, and bulls [9,10,11,12,13]. For example, fattening pig diets supplemented with 0.12% GAA 60 days before slaughter improved lean meat yield and reduced backfat thickness [10]. Finishing gilts supplemented with 0.045% GAA improved meat quality by changing muscle fiber characteristics and reducing mandibular fat [11]. In addition, GAA was found to improve the carcass yield and levels of essential amino acids (EEAs) in the breast muscles of aged laying hens [12]. In ruminants, the administration of GAA (0.3, 0.6, or 0.9 g/kg) has the potential to improve body weight (BW), average daily gain (ADG), and feed efficiency in Angus bulls [14]. We had confirmed elevated insulin-like growth factor1 (IGF-1) in serum with the addition of GAA [15]. However, the binding of the IGF-1 and IGF-1 receptor can activate the mammalian target of rapamycin (mTOR), a master protein kinase involved in cellular growth and protein synthesis [16]. Therefore, we measured the relative expression related to the gene (PI3K, Akt1, mTOR, and P70S6K) in muscle, and explored the mechanism by which GAA promotes growth, especially protein deposition. However, Speer [17] noted that the ruminal degradation rate of GAA was 0.47–0.49 before it was absorbed into the lower digestive tract. Therefore, it is necessary to consider the efficacy of encapsulated GAA.

Studies have shown that different types of forage also affect lipid accumulation in Japanese Black steers [18], and increasing hay in the replacement of silage resulted in a great deal of lean growth potential in newly weaned beef steers [19]. In lambs, the increasing substitute of peanut vine hay with foxtail millet silage indeed improved growth performance and feed efficiency by increasing DMI [20]. Based on the studies above, it was hypothesized that the addition of different forms of GAA, specifically uncoated GAA (UGAA) or coated GAA (CGAA) in two forage-type diets, have the potential to improve growth and meat quality in rapid-growing lambs. To this end, the purpose of this study was conducted to examine the effects of dietary GAA supplementation on growth performance and carcass quality in rapid-growing lambs fed different forage types.

## 2. Materials and Methods

### 2.1. Guanidine Acetic Acid Products

In the present study, the UGAA products contained an effective GAA of no less than 984 g/kg. The CGAA products contained effective GAA of no less than 600 g/kg, with the main coating material being palm fat powder, the average rumen degradation rate of CGAA is 15%. Both GAA products were gifted by Heibei Guang Rui Co., Ltd., (Shijiazhuang, China). 

### 2.2. Experimental Design and Animal Feeding Management

Seventy-two male small-tailed *Han* lambs aged two months with similar initial body weights (BW = 12 ± 1.6 kg) served as experimental animals. After arrival in the experimental pens, all animals were numbered with ear tags, health checked, and injected with ivermectin to remove internal parasites before starting the feeding trial. Lambs were housed indoors, kept well ventilated, at a steady temperature, and a relative humidity through door and windows. In addition, kept lit at night.

A 2 × 3 factorial experimental feeding design was applied to the lambs, which were fed total mixed rations (TMR) with two forage types (oaten hay [OH] and oaten hay plus wheat silage [OHWS]) and three forms of added GAA per ration (control, basal ration without GAA; UGAA, the basal ration to which 1 g/kg of dry matter uncoated GAA product was added; and CGAA, the basal ration to which 1 g/kg of dry matter coated GAA product was added). Following the experimental design (2 × 3 = 6 treatments), each treatment (*n* = 12) was arranged across four pens with bamboo-slotted bedding, and each pen was randomly allocated three lambs. Taking into account environmental factors, all pens were randomly distributed. The feeding trial period consisted of Stage 1 (62 days, forage:concentrate = 25:75) and Stage 2 (58 days, forage:concentrate = 20:80). As shown in Table 1, all dietary rations at both stages were formulated to meet the nutrient requirement of 300 g gain/day, as recommended by the Nutrient Requirements of Small Ruminants [21]. All lambs at the pen level were fed the corresponding rations twice a day at 08:00 and 16:00, and the lambs were allowed ad libitum access to feed and water. The feed offered and refused at the pen level was recorded daily, and moisture content was determined once a week and used to calculate dry matter intake (DMI) throughout the experiment. Live BW (determined by a combined electronic livestock scale, patent number CN 20178801OU, maximum weight 500 kg, resolution 0.1 kg) at the beginning and end of each feeding period was recorded to calculate weight increments and average daily gain (ADG).

### 2.3. Sample Collection and Measurements

#### 2.3.1. Serum Index

On days 60 and 120, blood samples of all lambs were collected in the morning before feeding. Briefly, a 5 mL blood sample was collected from the jugular vein of lamb, and the blood was stored in sterile vacuum glass test tubes. The tubes were then centrifuged at 3000× *g* for 15 min. The serum samples were removed and stored at −20 °C to detect triglyceride (TG) and fatty acid synthase (FAS), acetyl coenzyme A carboxylase (ACC), and hormone-sensitive triglyceride lipase (HSL). 

#### 2.3.2. Carcass Characteristics and Meat Quality Routine Indices

At the end of the feeding experiment (day 121), all lambs were sacrificed after an 18 h fasting period by standard Halal procedures [22]. After removing the skin, head, gastrointestinal tract, and internal organs, individual hot carcass weight (HCW) was recorded immediately postmortem and before the carcasses were chilled at 4 °C in a cold chamber for 24 h. Ribeye area was calculated by making a cut between the 12th to 13th intercostal space and measuring the area using a grid with 1 cm × 1 cm squares. The pH of paired longissimus lumborum (LL) at 24 h postmortem was recorded by a pH meter on the following day. The pH meter had an automatic calibration at two points with a set of standard buffers (pH 7.01/10.01) and automatic temperature compensation from −5.0 to 105.0 °C. Water-holding capacity was determined by measuring the drip loss and cooking loss of the samples. Then, 1.5 cm of the LL was weighed (m_1_), the meat was hung in an inverted container using a wire, and the container was placed in a 4 °C refrigerator for 24 h. The samples were then reweighed (m_2_). The calculation formula used was as follows: drip loss = (m_1_ − m_2_)/m_1_ × 100% [23]. 

Cooking loss was determined as the percentage difference in weight between the pre- and post-cooked meat samples. A separate 1.5 cm section of the LL from each carcass was weighed (m_1_), placed in a Ziplock bag, and incubated for 1 h in a water bath maintained at 80 °C. The samples were removed from the water bath, cooled to room temperature, blotted dry, and weighed (m_2_). All samples were completed in one batch. Cooking loss was calculated as follows: cooking loss = (m_1_ − m_2_)/m_1_ × 100% [23]. Meat color was scanned after a 30-min blooming period using a spectrophotometric colorimeter (CR-400 Chromameter, Konica Minolta, Sydney, NSW, Australia) fitted with a standard xenon lamp and an 8 mm aperture and set to Illuminant D65 and 2° standard observer. The instrument is calibrated with a white standard board. Lightness (L*), redness (a*), and yellowness (b*) were recorded three times, as described by Biffin et al. [24]. Backfat thickness was measured with a digital caliper at a point 4 cm from the carcass midline and 4 cm from the caudal edge of the last rib [25]. The LL was collected and freeze dried for evaluation. The methodology for meat and meat products described by Association of Official Analytical Chemists (AOAC) [26] was used for the analysis of the moisture (9341.01), ash (924.05), crude protein (CP) (920.87), and ether extract (EE) content (920.85). Briefly, the moisture content was measured by oven-drying the samples to a constant weight at 105 °C. The CP (N × 6.25) and EE contents were determined using the Kjeldahl method and the Soxhlet extraction method, respectively.

Amino acids were assayed using a Hitachi L-8900 instrument (Hitachi, Tokyo, Japan). Then, 50–60 mg dried meat samples were weighed accurately and then hydrolyzed with 6 mol/L hydrochloric acid (10 mL) for 24 h at 110 °C in a vacuum. After cooling, the solution was kept at a constant volume, filtered, and evaporated. Then, roughly 3 mL of sodium citrate buffer solution was added and centrifuged with the supernatant for analysis. The levels of 17 amino acids (aspartic [Asp], threonine [Thr], serine [Ser], glutamic [Glu], glycine [Gly], alanine [Ala], cysteine [Cys], valine [Val], methionine [Met], isoleucine [Ile], leucine [Leu], tyrosine [Tyr], phenylalanine [Phe], lysine [Lys], histidine [His], arginine [Arg], and proline [Pro]) in LL were analyzed.

#### 2.3.3. Longissimus Lumborum Gene Expression

The muscle samples were collected from the same sites and stored in a −80 °C environment for further RNA extraction. Total RNA was extracted from 50–100 mg of muscle tissue using the total RNA extraction reagent (Baifeite Bio Inc., Chengdu, China) following the manufacturer’s protocol. After RNA extraction, RNA was purified using the miRNeasy Mini Kit (Qiagen, Valencia, CA, USA), followed by on-column digestion with the RNase-free DNase (Qiagen, Valencia, CA, USA) RNA was quantified by NanoDrop ND-1000 spectrophotometer (NanoDrop Technologies, Wilmington, DE, USA). The A260/A280 ratio was 1.8 to 2.1 for all samples. RNA integrity was confirmed by denaturing 1.5% agarose gel electrophoresis. The cDNA was prepared by using Goldenstar RT6 cDNA synthesis kit version 2 (TSK302S, TSING KE Bio Inc., Wuhan, China) according to the manufacturer’s instructions. Relative expression of PIK3C3, Akt1, mechanistic target of rapamycin (mTOR), and MaFBx was quantified by real-time polymerase chain reaction (PCR) using the iCycler and iQ-SYBR Green Detection Kit (BioRad Laboratories, Hercules, CA, USA). The primer sets used for real-time PCR are listed in Table 2. Subsequent quantitative (q) PCR was performed on FDQ-96A multiplex qPCR systems (Hangzhoubori, Hangzhou, China) at a minimum in triplicate. A reaction mixture (20 μL) containing 1.0 μL cDNA, 10.0 μL SYBR Green I (TSING KE Bio Inc., Wuhan, China), 1.0 μL PCR forward primer (10 μM), 1.0 μL PCR reverse primer (10 μM), and 7.0 μL ddH_2_O was prepared. PCR was performed under the following cycle conditions: 1 cycle at 95 °C for 1 min, 40 cycles at 95 °C for 15 s, 60 °C for 15 s, and 72 °C for 30 s, respectively, as well as annealing temperature at 1 cycle of 95 °C for 5 s, 1 cycle of 60 °C for 1 min, 1 cycle of 0.11 °C/s temperature increase to 95 °C, 1 cycle of 50 °C for 30 s, followed by melting. At the end of each denaturation and extension step, fluorescence detection was carried out. In this study, the housekeeping gene β-actin was used as a reference index and for normalization. Relative mRNA levels were then calculated for each gene using ΔΔCT (threshold cycle) method. CT of housekeeping genes was subtracted from CT of each gene to obtain ΔCT. Calibrator ΔCT was subtracted from each sample’s ΔCT, and then relative mRNA value was calculated by the formula 2^−ΔΔCT^ [27].

### 2.4. Statistical Analysis

All data were analyzed in a completely randomized design with a factorial arrangement using SAS Systems software [28]. Growth, slaughter performance, enzymes associated with fat synthesis or degradation, meat traits, and muscle gene expression were analyzed using the MIXED procedure. The model was applied as follows:$$Y_{ijk}=\mu +G_{i}+F_{j}+{\left(G\times F\right)}_{ij}+R_{k}+eijk$$
where *Y_ijk_* is the dependent variable, *µ* is the overall mean, *G_i_* is the fixed effect of GAA products (*i* = 3: control, uncoated GAA, and coated GAA), *F_j_* is the fixed effect of the total mixed ration with different forage type (OH or OHWS), and *G* × *F* is the interaction between forage type and GAA. *R_k_* is the random effect of animals or pens, and *eijk* is a residual error term. The least square means and standard errors of the means were calculated with the LSMEANS statement of the SAS software (SAS Institute Inc., Cary, NC, USA). Significance was declared at *p* ≤ 0.05 unless otherwise noted.

## 3. Results

### 3.1. Growth and Slaughter Performance

Forage × GAA had no significant effects on BW, ADG, DMI, and F:G among all the groups (Table 3). The lambs in the OH group presented greater final BW (*p* = 0.031), DMI (*p* = 0.010), and ADG (*p* = 0.022) compared with the OHWS group. In addition, whatever form of GAA was added, the lambs presented greater final BW (*p* = 0.030) and ADG (*p* = 0.014), with only somewhat numerical increments in DMI (*p* = 0.058). The overall F:G did not alter in response to either forage type or added GAA (*p* = 0.500 or *p* = 0.359). 

The forage type × GAA interaction did not have a significant impact on HCW, dressing percentage, ribeye area, and backfat thickness. As shown in Table 4, compared with the OHWS diet, the lambs fed the OH diet had greater HCW (*p* = 0.001). The dressing percentage, ribeye area, and backfat thickness were similar among the two forage diets. Regardless of the form of GAA added, the lambs presented greater HCW (*p* < 0.01), dressing percentage (*p* < 0.001), and ribeye area (*p* < 0.001) but lower backfat thickness (*p* = 0.003). 

### 3.2. Serum Index

As shown in Table 5, the forage × GAA interaction was not observed for serum TG, FAS, ACC, or HSL at Stage 1 and Stage 2.

At Stage 1, no differences were found for serum TG, FAS, ACC, or HSL in the two forage-type diets. Dietary supplementation with UGAA and CGAA significantly decreased the concentrations of TG (*p* < 0.001) and increased (*p* < 0.001) the activity of the HSL enzyme.

At Stage 2, the TG content, FAS, ACC, or HSL activity were not correlated with the two forage-type diets. Regardless of the form of GAA added, the lambs presented greater (*p* < 0.001) HSL activity and lower (*p* < 0.001) serum TG. 

### 3.3. Meat Quality Traits

As shown in Table 6, forage × GAA had no significant effects on serum pH, meat color, water-holding capacity, or nutrient elements. In addition, meat quality and nutrient elements were not significantly correlated with the two forage type diets. The addition of UGAA or CGAA decreased drip loss (*p* = 0.001) and cooking loss (*p* = 0.001) but increased the pH value (*p* = 0.003) and CP content (*p* < 0.001) in mutton. 

The measured amino acid levels in the muscle tissues are presented in Table 7. Interaction between forage type and GAA addition was found for the non-essential amino acids (NEAAs) and delicious amino acids (DAAs). The addition of UGAA or CGAA increased the NEAAs and DAAs content (*p* < 0.05). CGAA addition in the OH group increased NEAAs and DAAs more than UGAA addition (*p* < 0.05), while there was no difference in the OHWS group (*p* > 0.05). Compared with the OHWS diet, the lambs fed the OH diet had lower TAAs (*p* = 0.001), EAAs (*p* = 0.021), NEAAs (*p* = 0.003), and DAAs (*p* = 0.004). Compared with the control, a significant increase in Asp (*p* < 0.001), Thr (*p* < 0.001), Ser (*p* = 0.015), Glu (*p* < 0.001), Pro (*p* < 0.001), Gly (*p* < 0.001), Ala (*p* < 0.001), Met (*p* < 0.001), Ile (*p* < 0.001), Leu (*p* < 0.001), Phe (*p* = 0.001), Lys (*p* < 0.001), His (*p* = 0.003), Arg (*p* = 0.001), TAAs (*p* < 0.001), EAAs (*p* < 0.001), NEAAs (*p* < 0.001), DAAs (*p* < 0.001), and branched chain amino acids (BCAAs) (*p* < 0.001) were obtained with the addition of GAA.

### 3.4. Muscle Gene Expression

The forage × GAA interaction did not have a significant impact on the mRNA levels of PIK3C3, mTOR, Akt1, and MaFBx (Table 8). There was no variation in muscle gene expression in the lambs fed the two different forage types. However, the mRNA levels of PIK3C3 (*p* < 0.001), mTOR (*p* < 0.001) and Akt1 (*p* < 0.001) were upregulated with the supplementation of UGAA or CGAA in the two diets. However, the expression of the MaFBx gene was down-regulated (*p* = 0.002) with GAA addition.

## 4. Discussion

### 4.1. Growth and Slaughter Performance

In this study, lambs fed the OH diet tended to grow faster (ADG > 300 g/d), as evidenced by the greater final BW, DMI, and ADG compared with the lambs fed the OHWS diet. This result also confirmed that silage (DM) can replace equal amounts of hay to save TMR but reduce ADG [18]. However, the F:G is not affected by the forage type in the current study. The possible reason was the different ratio of silage and hay, as well as the inconsistent ratio of concentrate and forage.

At present, GAA has been used as a safe feed additive to improve growth performance and meat quality in monogastric animals [29,30]. In cattle, the administration of GAA (0.3, 0.6, or 0.9 g/kg) has the potential to improve BW, ADG, and feed efficiency in Angus bulls [14,31]. Consistent with trial expectations, lambs showed higher final BW and ADG regardless of the form of GAA added. However, no improvements in feed efficiency were observed in this trial, this may be related to the amount of GAA. In our study, the interaction between forage type and GAA addition was not found for growth performance; this also showed that the positive effect of GAA on lamb growth performance was not affected by forage type.

The current trial resulted in a higher HCW for lambs on the OH diet compared with those on the OHWS diet. This was also evidenced by the increased carcass weight and decreased fat ratio following a spineless cactus diet associated with Tifton hay as a replacement for corn silage in sheep diets [32]. However, due to the limitations of the conditions, the untested fat ratio and lean meat rate were also shortcomings of this study, as these factors could only be indirectly reflected by backfat thickness and ribeye area. According to Kim et al. [33], different forage sources can regulate the differentiation of adipocytes in the sheep body and influence the adiposity of the carcasses. Unfortunately, the two forage diets showed no effects on backfat thickness and ribeye area. A possible reason is that the proportion of whole wheat silage in the OHWS group was too low, which had little effect on the CP and metabolic energy (ME) of the diet, which could affect backfat thickness and ribeye area [34].

Moreover, 0.3 g/kg GAA improves the HCW in growing-finishing pigs [35]. In the present study, we found that feeding lambs with UGAA or CGAA in two forage-type diets increased HCW and dressing percentage for rapid-growing lambs. Furthermore, previous studies have shown that supplementation with GAA may reduce abdominal fat (1 g/kg diet on broiler chicken) [36] and backfat thickness (1.2 g/kg diet on pigs) [10]. In pigs, the addition of GAA (300 mg/kg, 600 mg/kg, and 900 mg/kg diet) increased the ribeye area [29]. Moreover, 0.09% GAA was found to improve the ribeye area in lambs [37], these were consistent with our finding that the addition of 1 g/kg UGAA or CGAA improve the ribeye area in lambs. Studies have shown a strong correlation between ribeye area and the meat production performance of livestock [16]. In addition, we found that feeding lambs with GAA led to lower backfat thickness, which is consistent with a previous report on pigs [29]. From the available data, GAA improved carcass production and reduced fat deposition, but more indicators are needed to confirm this, such as carcass fat content, which was also a limitation of this study.

### 4.2. Serum Indexes

TG is a deposit of essential and non-essential fatty acids [38]. TG metabolism is composed of synthesis and catabolism. ACC and FAS are key enzymes of de novo lipogenesis, and HSL is an enzyme that catalyzes the rate-limiting hydrolysis step [39]. Previous studies have shown that replacing alfalfa hay with a mixture of cassava foliage silage and sweet potato vine silage did not affect TG [40]. In the present study, changes in TG levels and enzymes associated with fat metabolism were not observed in lambs with different roughage types. This could be attributed to the difference in fat content between the two forage diets being relatively small, and within the range of the lambs’ regulation. The present results show that the forage type did not affect the activity of the FAS, ACC, and HSL. This also explains the result of the similar backfat thickness obtained for the two forage types.

In the present study, GAA (UGAA or CGAA) addition decreased the serum TG in two stages. This was inconsistent with the study on pigs (0, 0.3, 0.6, 0.9 g/kg GAA) [28], which may be caused by different animal species. In addition, there was little change in serum ACC and FAS activity. This is inconsistent with Lu et al. [28], who found that the addition of GAA to pig diets reduced liver FAS activity, possibly due to differences in the site of detection. ACC is an essential rate-limiting enzyme in fatty acid synthesis [41], the results of this trial showed that the dosage of GAA had little effect on fatty acid synthesis. Regardless of the form of GAA added, the lambs presented greater HSL activity. We speculated that GAA is not related to fat synthesis. In addition, the elevated serum HSL activity in this study illustrated a decrease in fat deposition and a decrease in TG content maybe due to increased fat degradation, which also corresponds to a decrease in backfat thickness (Table 4) as a result of added UGAA or CGAA.

### 4.3. Meat Quality Traits

Júnior et al. [42] reported that the weight at slaughter influences the attributes of sheep meat. In the present study, we found no effect on pH, meat color, or water-holding capacity by forage type, this concurs with a previous study that compared maize silage and peanut hay in which respect [43]. The nutritional composition of meat is a significant determinant of lamb eating quality. Previous studies [32,34] found that moisture, CP, ash, and EE were not affected by forage type. Similar results were found in this study. In addition, consistent with prior studies on beef cattle [44], we found that diet forage type had little effect on individual amino acid composition. However, compared with the OHWS diet, the OH diet had lower TAAs, EAAs, NEAAs, and DAAs; this may be due to the different amino acid content in the two diets, which remains to be further proven.

In the present study, the muscle pH at 24 h of the UGAA or CGAA groups was higher than those of the control, indicating that the dietary addition of GAA could attenuate the rapid decline of muscle pH. This had also been confirmed in previous studies [25,37]. Meat color is related to the oxidation state of pigment proteins, although we had proved that GAA addition improved the antioxidant capacity [15]. The meat color was similar among the two forage types and the forms of GAA added, which was consistent with a recent report in lamb [37]. One of the possible reasons for this discrepancy was the dark color of sheep meat, which masked color improvement. Furthermore, the addition of UGAA or CGAA decreased cooking loss and the drip loss of muscle, possibly because GAA supplementation increased the content of creatine and phosphocreatine in the muscle cells, which increased the osmotic pressure of muscle cells. Accordingly, the muscle cells could hold water better, thereby reducing drip loss and cooking loss [29,31].

In a previous study, the addition of 0.09% GAA to the diet increased the CP content in the LL of Dorper (
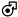
 ) × Small Tailed Han sheep (
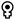
) crossed ram lambs [37]. Meanwhile dietary creatine supplementation enhanced the retention of muscle proteins [45]. Similar findings were obtained in this study with 1 g/kg UGAA or CGAA in two forage diets. 

Another prior study demonstrated that GAA supplementation increased the levels of Leu, Phe, Thr, Met, Arg, Glu, Pro, and His in the breast and thigh muscles of aged laying hens [12]. The present study demonstrated that the content of amino acids (Asp, Thr, Ser, Glu, Pro, Gly, Ala, Met, Ile, Leu, Phe, Lys, His, and Arg) and TAAs, EAAs, NEAAs, DAAs, BCAAs increased with UGAA or CGAA supplementation. Furthermore, this study also demonstrated an increase in CP content with the addition of GAA, as shown in Table 7. The exact change in AAs depended on the animal breed and the amount of GAA added in the present case.

In another vein, prior studies have indicated that EAAs content is an important factor in determining the nutritional value of meat [46]. Xiao et al. [47] also reviewed the effects of EAAs deficiency on animal BW, DMI, and energy metabolism. In our study, the addition of both forms of GAA to either diet significantly increased the EAA content, also indicating that the addition of GAA has a positive effect on meat quality and lamb growth. BCAAs are important nutritional signals, which provide the basis for protein synthesis and energy production [48]. In addition, BCAAs exert responses on several cell signaling mechanisms, mainly through the activation of the mTOR axis (the central regulator of mammalian cell growth) [49]. Our results showed that the addition of UGAA or CGAA to both diets increased the content of BCAAs and provided a direct basis for the addition of GAA to improve protein synthesis.

DAAs are important factors that affect flavor [50] and contribute to the taste of many foods [51]. In particular, Glu in DAAs is considered the amino acid with the greatest impact on meat flavor [52]. Previous studies have shown that the dietary addition of GAA leads to spared Arg, causing more Arg to participate in protein synthesis [5]. Studies on fish showed that TAAs and DAAs content increased with creatine supplementation [45]. This is consistent with the increase in Arg, TAAs, and DAAs content with the addition of UGAA or CGAA in the present study. In addition, Arg has been found to differentially regulate the expression of fat metabolic genes in skeletal muscle and white adipose tissue, therefore favoring lipogenesis in muscle but lipolysis in adipose tissue [53]. This also corresponds to serum indicators of fat metabolism. In the present study, compared with the control group, diets supplemented with UGAA or CGAA increased the contents of DAAs. This also indicates that the addition of UGAA or CGAA to the two diets had a flavor-enhancing effect under the conditions of this study. From the meat quality assessment results, dietary GAA improved the overall quality of the lamb and the CP deposited in the mutton (mainly due to an increase in AA content). Under the conditions of this test, the effect of CGAA was more obvious. A possible explanation is that a portion of UGAA was broken down as it passed through the rumen, resulting in a lower amount of muscle tissue than the CCAA.

### 4.4. Musical Gene Expression

The mTOR is a nutrient sensor and master regulator of metabolism that integrates hormone- and growth factor-induced growth signals to stimulate anabolic, and inhibit catabolic, processes. Akt signaling stimulates protein synthesis by activating mTOR and inhibits protein degradation by repressing the transcription factor FoxO, leading to the regulation of MaFBx/atrogin-1 and MuRF1 expression [54].

Differences in muscle gene expression were also not found to be a result of the diets of the two forage types, and there could be two reasons, one was the low proportion of wheat silage substitution, and the other was the limited number of genes detected.

The increase in muscle mRNA expressions of PIK3C3, mTOR, and Akt1 with UGAA or CGAA addition was in line with the findings of Liu et al. [13], who reported that the addition of 0.6 g/kg GAA in Angus bulls upregulated the PIK3C3, mTOR, Akt1, and P70S6K. Wang et al. [55] reported that GAA supplementation could promote myoblast differentiation and skeletal muscle growth through the miR-133a-3p- and miR-1a-3p-induced activation of the AKT/mTOR/S6K signaling pathways. Akt/mTOR/S6K signaling is a typical protein synthesis pathway. These results suggest that the observed increase in meat CP might be associated with the regulation of UGAA or CGAA to the expression of genes in the Akt1, PIK3C3, and mTOR pathways. MAFbx was termed ‘atrogenes’, which is positively correlated with protein degradation [56]. In the present study, Akt1, PIK3C3, and mTOR expression were upregulated and the MaFBx genes was down-regulated in the muscle with the addition of UGAA or CGAA, which indicated that the addition of GAA had the potential to improve protein deposition.

## 5. Conclusions

Under a high-concentrate diet, the lambs fed either form of GAA exhibited rapid growth (ADG > 300 g/d). Both forms of GAA presented greater final BW, improved carcass quality with less backfat thickness, and a higher water-holding capacity. GAA addition increased the meat protein mass and AA content in mutton by regulating the Akt1, PIK3C3, mTOR, and MaFBx expression. Collectively, these results suggested that GAA might be nutritional strategies to improve growth performance, carcass traits, and meat quality.

## Figures and Tables

**Table 1 foods-12-00641-t001:** Ingredient and chemical composition of experimental diets for lambs.

Ingredients	Stage 1		Stage 2	
	OH	OHWS	OH	OHWS
Wheat silage ^1^	0.00	90.00	0.00	70.00
Oaten hay ^2^	250.00	160.00	200.00	130.00
Concentrate ^3^	750.00	750.00	800.00	800.00
Nutrient level for TMR (g/kg, as DM)				
Organic matter	893.31	901.81	905.63	910.51
Crude protein	191.91	193.74	184.58	174.52
Ether extract	36.63	35.13	38.02	37.53
Neutral detergent fiber	307.61	293.79	274.51	271.24
Acid detergent fiber	118.41	117.12	108.00	99.72
Calcium	0.65	0.65	0.62	0.62
Phosphorous	0.39	0.38	0.37	0.37
Gross energy (MJ/kg)	15.46	15.51	17.03	16.84
Metabolic energy (MJ/kg)	8.07	8.10	8.34	8.39

^1^ Wheat silage: DM 879.5 g/kg; CP 121.0 g/kg; EE 24.1 g/kg; NDF 520.0 g/kg; ADF 310.5 g/kg; OM 917.0 g/kg; GE 14.4 MJ/kg. ^2^ Oaten hay: DM 329.3 g/kg; CP 120.8 g/kg; EE 32.0 g/kg; NDF 620.0 g/kg; ADF 370.5 g/kg; OM 919.0 g/kg; GE 14.2 MJ/kg. ^3^ Contained per kg in Stage1: 380 g Corn meal, 150 g Soybean meal, 180 g DDGS, 40 g Premix, Contained per kg in Stage2: 500 g Corn meal, 150 g Soybean meal, 110 g DDGS, 40 g Premix. Contained per kg premix: vitamin A, 150,000 IU; vitamin D_3_, 50,000 IU; vitamin E, 500 IU, vitamin B_1_, 200 IU; Fe, 1800 mg; Mn, 1500 mg; Zn, 1000 mg; Cu 350 mg I, 10.0 mg; Se, 3 mg; Co, 5 mg; Ca, 100 g; NaCl, 100 g; Total P, 3 g.

**Table 2 foods-12-00641-t002:** PCR primers for real-time PCR assay.

Genes	Primer Sequence	GenBank Accession No.	Size (bp)
PIK3C3	Fwd-5′-GAGACGGTTCAGCCAAGCAT-3′ Rev-5′-TTTCCACTTTCACGCTGCAC-3′	NC_056076.1	132
Akt1	Fwd-5′-AGTACATCAAGACCTGGCGG-3′ Rev-5′-GAGAAGTTGTTGAGGGGCGA-3′	FJ_943991.1	118
mTOR	Fwd-5′-GAACCCAGCCTTTGTCATGC-3′ Rev-5′-GGGCACTCTGCTCCTTGATT-3′	FJ_617140.1	101
MaFBx	Fwd-5′-TGGTCCAAAGAGTCGGCAAG-3′ Rev-5′-AAGCACAAAGGCAGGTCTGT-3′	XM_004011657.5	157
β-actin	Fwd-5′-GCACCCAGCACGATGAAGAT-3′ Rev-5′-AAGAAAAAGGGTGTAACGCAGC-3′	U_39357.2	201

PIK3C3, phosphoinositide 3-kinase; Akt1, protein kinase B; Mtor, mammalian target of rapamycin; MaFBx, muscle atrophy F-box.

**Table 3 foods-12-00641-t003:** Effects of forage type and GAA addition on growth performance in lambs.

Items	Forage	GAA Addition			SEM	*p*-Value		
		Control	UGAA	CGAA		Forage	GAA	Forage × GAA
Initial BW (kg)	OH	13.03	13.24	12.95	0.57	0.806	0.888	0.975
	OHWS	13.06	13.04	12.77				
Final BW (kg)	OH	47.45 ^b^	49.83 ^ab^	51.09 ^a^	1.03	0.031	0.030	0.494
	OHWS	45.66 ^b^	48.91 ^a^	47.41 ^ab^				
DMI (g/day)	OH	1116.95 ^b^	1237.12 ^ab^	1240.75 ^a^	22.20	0.010	0.058	0.358
	OHWS	1150.17 ^b^	1180.09 ^a^	1162.86 ^b^				
ADG (g)	OH	286.91 ^b^	301.51 ^ab^	317.86 ^a^	8.00	0.022	0.014	0.358
	OHWS	272.58 ^b^	296.63 ^a^	288.65 ^ab^				
F:G	OH	4.08	4.10	3.93	0.12	0.500	0.359	0.537
	OHWS	4.26	4.00	4.05				

BW, body weight; DMI, dry matter intake; ADG, average daily gain; F:G, feed-to-gain ratio calculated as DMI divided by ADG, SEM, standard error of least square means. ^a,b^ means with different superscripts were significantly different (*p* < 0.05).

**Table 4 foods-12-00641-t004:** Effects of forage type and GAA addition on slaughter performance in lambs.

Items	Forage	GAA Addition			SEM	*p*-Value		
		Control	UGAA	CGAA		Forage	GAA	Forage × GAA
Hot carcass weight (kg)	OH	22.74 ^b^	24.69 ^a^	25.46 ^a^	0.29	0.001	<0.001	0.097
	OHWS	22.09 ^b^	24.34 ^a^	23.87 ^ab^				
Dressing percentage (%)	OH	47.57 ^b^	49.12 ^a^	49.15 ^a^	0.26	0.519	<0.001	0.080
	OHWS	47.03 ^b^	49.61 ^a^	49.62 ^a^				
Backfat thickness (cm)	OH	7.28 ^a^	7.16 ^a^	6.95 ^b^	0.10	0.449	0.003	0.292
	OHWS	7.33 ^a^	6.91 ^b^	6.95 ^ab^				
Ribeye area (cm^2^)	OH	25.25 ^b^	28.33 ^a^	29.08 ^a^	0.40	0.836	<0.001	0.087
	OHWS	25.67 ^c^	29.08 ^a^	28.12 ^b^				

SEM, standard error of least square means. ^a,b,c^ means with different superscripts were significantly different (*p* < 0.05).

**Table 5 foods-12-00641-t005:** Effects of forage type and GAA addition on serum indexes in feedlotting lambs.

Items	Forage	GAA Addition			SEM	*p*-Value		
		Control	UGAA	CGAA		Forage	GAA	Forage × GAA
Stage 1 (60 d)								
TG (mmol/L)	OH	0.44 ^a^	0.40 ^b^	0.40 ^b^	0.01	0.333	<0.001	0.873
	OHWS	0.45 ^a^	0.41 ^b^	0.41 ^b^				
FAS (ng/mL)	OH	3.68	3.70	3.74	0.03	0.655	0.644	0.393
	OHWS	3.70	3.71	3.69				
ACC (ng/mL)	OH	30.37	31.57	31.21	0.46	0.876	0.693	0.191
	OHWS	31.35	30.90	31.08				
HSL (ng/mL)	OH	1.40 ^c^	1.50 ^b^	1.56 ^a^	0.02	0.501	<0.001	0.743
	OHWS	1.42 ^b^	1.51 ^a^	1.55 ^ab^				
Stage 2 (120 d)								
TG (mmol/L)	OH	0.50 ^a^	0.42 ^b^	0.45 ^b^	0.01	0.918	<0.001	0.171
	OHWS	0.48 ^a^	0.45 ^b^	0.44 ^b^				
FAS (ng/mL)	OH	3.48	3.53	3.49	0.06	0.312	0.440	0.303
	OHWS	3.40	3.40	3.55				
ACC (ng/mL)	OH	27.47	28.71	28.34	0.61	0.616	0.401	0.723
	OHWS	27.79	28.22	27.76				
HSL (ng/mL)	OH	1.34 ^b^	1.48 ^a^	1.52 ^a^	0.01	0.715	<0.001	0.596
	OHWS	1.35 ^b^	1.48 ^a^	1.50 ^a^				

TG, triglyceride; FAS, fatty acid synthase; ACC, acetyl coenzyme A carboxylase; HSL, hormone sensitive triglyceride lipase; SEM, standard error of least square means. ^a,b,c^ means with different superscripts were significantly different (*p* < 0.05).

**Table 6 foods-12-00641-t006:** Effects of forage type and GAA addition on meat quality traits in feedlotting lambs.

Items	Forage	GAA Addition			SEM	*p*-Value		
		Control	UGAA	CGAA		Forage	GAA	Forage × GAA
pH	OH	5.46	5.59	5.58	0.04	0.174	0.003	0.860
	OHWS	5.42 ^b^	5.52 ^a^	5.56 ^a^				
Meat color								
L*	OH	30.65	30.70	29.85	0.57	0.258	0.341	0.948
	OHWS	30.98	31.30	30.53				
a*	OH	16.14	16.08	16.05	0.26	0.810	0.154	0.111
	OHWS	15.66	16.71	16.06				
b*	OH	8.29	8.26	8.12	0.19	0.629	0.649	0.805
	OHWS	8.28	8.03	8.12				
Water-holding capacity								
Drip loss (%)	OH	20.21 ^a^	19.72 ^ab^	19.19 ^b^	0.30	0.786	0.001	0.083
	OHWS	20.86 ^a^	19.04 ^b^	19.42 ^b^				
Cooking loss (%)	OH	45.04 ^a^	43.54 ^b^	42.91 ^b^	0.37	0.389	0.001	0.209
	OHWS	45.17 ^a^	42.51 ^b^	43.03 ^b^				
Nutrient elements (%)								
Moisture	OH	76.01	75.93	75.91	0.17	0.743	0.792	0.854
	OHWS	75.91	75.99	75.80				
Crude Protein	OH	20.07 ^b^	20.63 ^a^	20.94 ^a^	0.13	0.502	<0.001	0.075
	OHWS	20.09 ^b^	20.81 ^a^	20.53 ^ab^				
Ether extract	OH	2.10	2.10	2.23	0.09	0.904	0.980	0.308
	OHWS	2.16	2.18	2.06				
Ash	OH	1.16	1.21	1.23	0.03	0.724	0.308	0.753
	OHWS	1.18	1.20	1.20				

SEM, standard error of least square means. ^a,b^ means with different superscripts were significantly different (*p* < 0.05).

**Table 7 foods-12-00641-t007:** Effects of forage type and GAA addition on the amino acid composition in longissimus lumborum of feedlotting lambs.

Items	Forage	GAA Addition			SEM	*p*-Value		
		Control	UGAA	CGAA		Forage	GAA	Forage × GAA
Asp	OH	7.54 ^b^	7.79 ^a^	7.83 ^a^	0.03	0.187	<0.001	0.722
	OHWS	7.60 ^b^	7.78 ^a^	7.85 ^a^				
Thr	OH	3.50 ^c^	3.65 ^b^	3.73 ^a^	0.02	0.721	<0.001	0.954
	OHWS	3.50 ^c^	3.66 ^b^	3.74 ^a^				
Ser	OH	2.73	2.75	2.77	0.03	0.199	0.015	0.331
	OHWS	2.72 ^b^	2.78 ^a^	2.83 ^a^				
Glu	OH	14.36 ^b^	14.65 ^a^	14.69 ^a^	0.04	0.148	<0.001	0.332
	OHWS	14.40 ^b^	14.75 ^a^	14.67 ^a^				
Pro	OH	2.75 ^b^	3.88 ^a^	3.88 ^a^	0.03	0.703	<0.001	0.447
	OHWS	2.77 ^b^	3.89 ^a^	3.82 ^a^				
Gly	OH	3.61 ^b^	3.78 ^a^	3.83 ^a^	0.03	0.451	<0.001	0.404
	OHWS	3.63 ^b^	3.84 ^a^	3.81 ^a^				
Ala	OH	4.52 ^b^	4.66 ^a^	4.68 ^a^	0.02	0.057	<0.001	0.493
	OHWS	4.53 ^b^	4.72 ^a^	4.73 ^a^				
Cys	OH	0.72	0.74	0.75	0.02	0.299	0.223	0.708
	OHWS	0.73	0.77	0.75				
Val	OH	4.33	4.37	4.37	0.03	0.588	0.114	0.724
	OHWS	4.34	4.40	4.36				
Met	OH	2.27 ^b^	2.34 ^a^	2.35 ^a^	0.02	0.301	<0.001	0.339
	OHWS	2.30 ^b^	2.36 ^a^	2.34 ^ab^				
Ile	OH	4.14 ^b^	4.26 ^a^	4.26 ^a^	0.01	0.943	<0.001	0.848
	OHWS	4.15 ^b^	4.25 ^a^	4.27 ^a^				
Leu	OH	6.66 ^b^	6.68 ^b^	6.76 ^a^	0.02	0.443	<0.001	0.911
	OHWS	6.67 ^b^	6.68 ^b^	6.78 ^a^				
Tyr	OH	2.81 ^b^	2.93 ^a^	2.94 ^a^	0.05	0.689	0.069	0.928
	OHWS	2.85	2.94	2.94				
Phe	OH	4.01 ^b^	4.17 ^a^	4.21 ^a^	0.06	0.244	0.001	0.192
	OHWS	4.08 ^b^	4.11 ^b^	4.36 ^a^				
Lys	OH	7.19 ^b^	7.28 ^a^	7.30 ^a^	0.02	0.395	<0.001	0.436
	OHWS	7.20 ^b^	7.32 ^a^	7.29 ^a^				
Trp	OH	1.32	1.33	1.34	0.02	0.551	0.618	0.716
	OHWS	1.33	1.35	1.34				
His	OH	2.72 ^b^	2.85 ^a^	2.89 ^a^	0.03	0.145	0.003	0.052
	OHWS	2.83	2.89	2.84				
Arg	OH	5.36 ^b^	5.44 ^a^	5.48 ^a^	0.03	0.942	0.001	0.978
	OHWS	5.36 ^b^	5.45 ^a^	5.47 ^a^				
TAAs	OH	80.53 ^c^	83.51 ^b^	84.04 ^a^	0.13	0.001	<0.001	0.148
	OHWS	80.96 ^b^	84.12 ^a^	84.16 ^a^				
EAAs	OH	40.18 ^c^	41.03 ^b^	41.34 ^a^	0.09	0.021	<0.001	0.650
	OHWS	40.43 ^b^	41.20 ^a^	41.43 ^a^				
NAAs	OH	39.03 ^c^	41.15 ^b^	41.36 ^a^	0.14	0.003	<0.001	0.044
	OHWS	39.20 ^b^	41.69 ^a^	41.39 ^a^				
DAAs	OH	37.66 ^c^	38.64 ^b^	38.86 ^a^	0.06	0.004	<0.001	0.039
	OHWS	37.82 ^b^	39.00 ^a^	38.84 ^a^				
BCAAs	OH	15.13 ^b^	15.31 ^a^	15.39 ^a^	0.04	0.169	<0.001	0.815
	OHWS	15.16 ^b^	15.38 ^a^	15.42 ^a^				

TAAs, total amino acids; EAAs, essential amino acids, EAAs = ∑(Thr + Val + Met + Ile + Leu + Phe + Lys + His + Arg); NEAAs, non-essential amino acids, NEAAs = ∑(Asp + Ser + Glu + Gly + Ala + Cys Tyr + Pro); DAAs, delicious amino acids, DAAs = ∑(Asp + Glu + Gly + Ala + Arg + Met); BCAAs, branched chain amino acids, BCAAs = ∑(Val + Ile + Leu); SEM, standard error of least square means. ^a,b,c^ means with different superscripts were significantly different (*p* < 0.05).

**Table 8 foods-12-00641-t008:** Effects of forage type and GAA addition on gene expression in feedlotting lambs.

Items	Forage	GAA Addition			SEM	*p*-Value		
		Control	UGAA	CGAA		Forage	GAA	Forage × GAA
PIK3C3	OH	0.84 ^b^	1.19 ^a^	1.18 ^a^	0.02	0.762	<0.001	0.832
	OHWS	0.83 ^b^	1.19 ^a^	1.17 ^a^				
mTOR	OH	0.67 ^b^	1.03 ^a^	1.01 ^a^	0.02	0.964	<0.001	0.745
	OHWS	0.65 ^b^	1.03 ^a^	1.02 ^a^				
Akt1	OH	0.93 ^b^	1.13 ^a^	1.12 ^a^	0.05	0.568	<0.001	0.984
	OHWS	0.92 ^b^	1.11 ^a^	1.11 ^a^				
MaFBx	OH	1.10 ^a^	1.06 ^ab^	1.00 ^b^	0.03	0.963	0.002	0.914
	OHWS	1.11 ^a^	1.05 ^ab^	1.00 ^b^				

Akt1, protein kinase B; MaFBx, muscle atrophy F-box; mTOR, mammalian target of rapamycin; PIK3C3, phosphoinositide 3-kinase; SEM, standard error of least square means. ^a,b^ means with different superscripts were significantly different (*p* < 0.05).

## Data Availability

Not applicable.

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
