# Peer review of "Dietary Guanidine Acetic Acid Addition Improved Carcass Quality with Less Back-Fat Thickness and Remarkably Increased Meat Protein Deposition in Rapid-Growing Lambs Fed Different Forage Types"

_foods, 2023, doi:10.3390/foods12030641_

Round 1

Reviewer 1 Report

Title: Dietary guanidine acetic acid addition...different types of forage

It sounds better.

Abstract: Mention which types of meat quality were tested.

Introduction:

Page 1, line 38- Hence, a question arises as to how to shorten the feeding cycle and increase the rapid-growing lambs meat production? Change the punctuation.

Materials and Methods:

Page 3, Table 1, Nutrient level (g/kg, as DM) for what type of feed?

Page 3, line 122, Why the authors used Halal procedures? Hide should be replaced with skin

Result

Page 6, line 216,  As shown in Table 4. Compared to…  Should be corrected.

Table 4. hot carcass weight, 1st letter uppercase

Discussion: Is it cost effective? Is there any report?

Author Response

Point 1: Page 1, line 38- Hence, a question arises as to how to shorten the feeding cycle and increase the rapid-growing lambs meat production? Change the punctuation.

Response 1: Yes, accepted and revised.

Point 2: Page 3, Table 1, Nutrient level (g/kg, as DM) for what type of feed?

Response 2: Yes, accepted and revised.

Point 3: Page 3, line 122, Why the authors used Halal procedures? Hide should be replaced with skin

Response 3: Yes, accepted and revised. Consider the local customs of the sheep farm,we used Halal procedures.

Point 4: Page 6, line 216,  As shown in Table 4. Compared to…  Should be corrected.

Response 4: Yes, accepted and revised.

Point 5:Table 4. hot carcass weight, 1st letter uppercase

Response 5: Yes, accepted and revised.

Point 6:Discussion: Is it cost effective? Is there any report?

Response 6: The focus of this study was carcass quality, the cost effective was not considered, which is a limitation of our study, and there have been no reports on GAA cost effective currently.

Reviewer 2 Report

The purpose of this study was to determine the effect of different forms of guanidine acetic acid (GAA) in two forage-type diets on growth performance and meat quality in rapid-growing lambs. The Introduction chapter provides an overview of the world's knowledge on this subject. The material used in the research is sufficiently numerous, but some supplementing the description in Materials and Methods chapter are needed. The results are described usually correctly. The discussion is exhaustive. Summary of the results are correct. Some corrections are needed. The proposed changes are listed below.

General comments:

Please, prepare the article in accordance with the instructions for the authors:

For significance, use a low letter "p" in italic instead of the capital "P" in the main article

The missing chapters: "Institutional Review Board Statement", and "Informed Consent Statement" must be added to article

In References chapter please use a "dot" after each abbreviation, for example „Anim. Sci. J.” instead of Anim Sci J

In Reference chapter for page ranges use long (-) from the symbol function, instead of short (-) from the keyboard

Enter the full page range, for example 165-171, 725-736, 929-937, etc instead of 165-71, 725-36, 929-37

Tables 1 to Table 8 not bold in a main text

Detailed Comments:

L18+ are the words "Background", "Results" "Conclusions" required? See other manuscripts in Foods journal

L24 greater or lower DMI in OH than OHWS diet?

L29 „decreased” the pH value? or the acidity of the meat? See Table 6

L73+ add the purpose of the study after the research hypothesis

L82 Were the animals kept in an enclosed building with no windows, no pastures? What was the temperature, relative humidity, lighting program?

L102 Give the name, type of scale, manufacturer's details, accuracy of measurement used to determine BW and feed intake (feed offered and refused)

L117 -20 °C. a space between the number and the unit of measure

L145 Please enter the data of the colorimeter calibration plate

L150 105 °C

L162 Space before a subsection

L163,180,181, 182, 183, etc space before °C

L203 please provide the name of the test to verify the significance of differences between the groups

L207 and other 'p' in italic and lower case

L211 after GAA add (p = 0.500 or p = 0.359)

L 216 "compared" in lower case

L212+ no interaction description

L220+ no interaction description

L226 delete (P > 0.191), there are 4 features

L229 add "at Stage 1"

L230 move to L233

L233 describe the interactions for Stage 1 and Stage 2 together

Author Response

Point 1: For significance, use a low letter "p" in italic instead of the capital "P" in the main article

Response 1: Yes, accepted and revised.

Point 2: The missing chapters: "Institutional Review Board Statement", and "Informed Consent Statement" must be added to article.

Response 2: Yes, accepted and revised.

Point 3: In Reference chapter for page ranges use long (-) from the symbol function, instead of short (-) from the keyboard

Response 3: Yes, accepted and revised.

Point 4: Enter the full page range, for example 165-171, 725-736, 929-937, etc instead of 165-71, 725-36, 929-37

Response 4: Yes, accepted and revised.

Point 5: Tables 1 to Table 8 not bold in a main text

Response 5: Yes, accepted and revised.

Point 6: L18+ are the words "Background", "Results" "Conclusions" required? See other manuscripts in Foods journal

Response 6: Yes, accepted and revised.

Point 7: L24 greater or lower DMI in OH than OHWS diet?

Response 7: As shown in Table 3, the OH diet had greater DMI than OHWS diet.

Point 8: L29 „decreased” the pH value? or the acidity of the meat? See Table 6

Response 8: Yes, accepted and revised. Dietary GAA decreased the acidity of the meat.

Point 9: L73+ add the purpose of the study after the research hypothesis

Response 9: Yes, accepted and revised.

Point 10: L82 Were the animals kept in an enclosed building with no windows, no pastures? What was the temperature, relative humidity, lighting program?

Response 10: Yes, accepted and rewrote the section.

Point 11: L102 Give the name, type of scale, manufacturer's details, accuracy of measurement used to determine BW and feed intake (feed offered and refused)

Response 11: Yes, accepted and revised.

Point 12: L117 -20 °C. a space between the number and the unit of measure

Response 12: Yes, accepted and revised.

Point 13: L145 Please enter the data of the colorimeter calibration plate

Response 13: Yes, accepted and revised. The instrument is calibrated with a white standard board.

Point 14: L150 105 °C

Response 14: Yes, accepted and revised.

Point 15: L162 Space before a subsection

Response 15: Yes, accepted and revised.

Point 16: L163,180,181, 182, 183, etc space before °C

Response 16: Yes, accepted and revised.

Point 17: L203 please provide the name of the test to verify the significance of differences between the groups

Response 17: Yes, accepted and revised.

Point 18: L207 and other 'p' in italic and lower case

Response 18: Yes, accepted and revised.

Point 19: L211 after GAA add (p = 0.500 or p = 0.359)

Response 19: Yes, accepted and revised.

Point 20: L 216 "compared" in lower case

Response 20: Yes, accepted and revised.

Point 21: L212+ no interaction description

Response 21: Yes, accepted and revised.

Point 22: L220+ no interaction description

Response 22: Yes, accepted and revised.

Point 23: L226 delete (P > 0.191), there are 4 features

Response 23: Yes, accepted and revised.

Point 24: L229 add "at Stage 1"

Response 24: Yes, accepted and revised.

Point 25: L230 move to L233

Response 25: Yes, accepted and revised.

Point 26: L233 describe the interactions for Stage 1 and Stage 2 together

Response 26: Yes, accepted and revised.

Reviewer 3 Report

Some adjustments needed to improve readability and comprehension.

-       Always report the unit of measurement in the same form. One space after the value, also for % and °C.

-       The layout of the table 3 and 5 in the pdf should be improved. Horizontal separating lines or empty row between items could be added

-       Observe the journal's instructions for references. No space between double name abbreviations.

-       Line 89, 90. 1 g/kg of Dry Matter or Wet Basis? 

-       Lines 90-92. Better specify the spatial distribution of all pens. Were the 24 pens randomly distributed in space? This is to randomly distribute the action of environmental factors (air streams, noise, etc.)

-       Line 117. “g” is the unit for gram. Use “rpm” unit.

-       Line 214. (feed:gain, g: g) ????? It is a ratio.

Author Response

Point 1: Always report the unit of measurement in the same form. One space after the value, also for % and °C.

Response 1: Yes, accepted and revised.

Point 2: The layout of the table 3 and 5 in the pdf should be improved. Horizontal separating lines or empty row between items could be added

Response 2: Yes, accepted and revised.

Point 3: Observe the journal's instructions for references. No space between double name abbreviations.

Response 3: Yes, accepted and revised.

Point 4: Line 89, 90. 1 g/kg of Dry Matter or Wet Basis?

Response 4: Yes, accepted and revised.

Point 5: Lines 90-92. Better specify the spatial distribution of all pens. Were the 24 pens randomly distributed in space? This is to randomly distribute the action of environmental factors (air streams, noise, etc.)

Response 5: Yes, accepted and revised. Taking into account environmental factors, all pens were randomly distributed.

Point 6: Line 117. “g” is the unit for gram. Use “rpm” unit.

Response 6: g” is the unit of “centrifugal force” in this study, and its magnitude depends on the centrifuge speed (rpm) and effective centrifugation radius (r).

Point 7: Line 214. (feed:gain, g: g) ????? It is a ratio.

Response 7: Yes, accepted and revised.
